# Peripheral Myelin Protein 22 Gene Mutations in Charcot-Marie-Tooth Disease Type 1E Patients

**DOI:** 10.3390/genes13071219

**Published:** 2022-07-08

**Authors:** Na Young Jung, Hye Mi Kwon, Da Eun Nam, Nasrin Tamanna, Ah Jin Lee, Sang Beom Kim, Byung-Ok Choi, Ki Wha Chung

**Affiliations:** 1Department of Biological Sciences, Kongju National University, Gongju 32588, Korea; jny7765@naver.com (N.Y.J.); daeun5612@naver.com (D.E.N.); emutamanna33@gmail.com (N.T.); jhmom1010@naver.com (A.J.L.); 2Department of Neurology, Samsung Medical Center, Sungkyunkwan University School of Medicine, Seoul 06351, Korea; huimei.kwon@samsung.com; 3Department of Neurology, Kyung Hee University Hospital at Gangdong, Kyung Hee University School of Medicine, Seoul 05278, Korea; sbkim@khu.ac.kr; 4Cell & Gene Therapy Institute, Samsung Medical Center, Seoul 06351, Korea

**Keywords:** Charcot-Marie-Tooth disease type 1E (CMT1E), Korean, *PMP22*, point mutation, whole-exome sequencing

## Abstract

Duplication and deletion of the peripheral myelin protein 22 (*PMP22*) gene cause Charcot-Marie-Tooth disease type 1A (CMT1A) and hereditary neuropathy with liability to pressure palsies (HNPP), respectively, while point mutations or small insertions and deletions (indels) usually cause CMT type 1E (CMT1E) or HNPP. This study was performed to identify *PMP22* mutations and to analyze the genotype–phenotype correlation in Korean CMT families. By the application of whole-exome sequencing (WES) and targeted gene panel sequencing (TS), we identified 14 pathogenic or likely pathogenic *PMP22* mutations in 21 families out of 850 CMT families who were negative for 17p12 (*PMP22*) duplication. Most mutations were located in the well-conserved transmembrane domains. Of these, eight mutations were not reported in other populations. High frequencies of de novo mutations were observed, and the mutation sites of c.68C>G and c.215C>T were suggested as the mutational hotspots. Affected individuals showed an early onset-severe phenotype and late onset-mild phenotype, and more than 40% of the CMT1E patients showed hearing loss. Physical and electrophysiological symptoms of the CMT1E patients were more severely damaged than those of CMT1A while similar to CMT1B caused by *MPZ* mutations. Our results will be useful for the reference data of Korean CMT1E and the molecular diagnosis of CMT1 with or without hearing loss.

## 1. Introduction

Charcot-Marie-Tooth disease (CMT), which is the most frequent disorder in the inherited peripheral neuropathies (IPNs), is a genetically and clinically heterogeneous disorder group characterized by distal muscle weakness and sensory loss. CMT is usually categorized into three types mainly according to the electrophysiological properties: demyelinating (CMT1), axonal (CMT2), and intermediate type (Int-CMT). CMT with genetic defects in X-linked genes is exceptionally called CMTX. However, each type is divided into variable subtypes according to the genetic causes and clinical features [1]. In a broad sense, CMT also commonly includes distal hereditary motor neuropathy (dHMN) and hereditary sensory autonomic neuropathy (HSAN).

In the past decade, next-generation sequencing (NGS) technology such as whole-exome sequencing (WES) has far improved the molecular diagnosis of CMT [2]. Mutations in more than 130 genes have been currently reported to cause CMT or CMT-related diseases. Of these, mutations in the peripheral myelin protein 22 (*PMP22*; MIM 601097) gene, which encodes a major component of myelin in the peripheral nervous system, provide the most frequent genetic causes of CMT. Duplication and deletion of a 1.4 Mbp-length 17p12 region including peripheral myelin protein 22 (*PMP22*; MIM 601097) cause CMT type 1A (CMT1A; MIM 118220) and hereditary neuropathy with liability to pressure palsies (HNPP; MIM 162500), respectively [3,4]. However, small insertion/deletions (indels) and point mutations in *PMP22* are usually associated with CMT1E (MIM 118300) [5,6,7] and HNPP [8]. They have been also reported to cause Dejerine-Sottas syndrome (DSS; MIM 145900) as a severe type of CMT [9,10,11,12,13] and chronic inflammatory demyelinating polyneuropathy (CIDP; MIM 139393) [14]. Some CMT1E patients frequently showed a severe clinical phenotype of wheelchair bounding within the first decade [15]. Most mutations in *PMP22* cause IPNs in a dominant manner with a minor exception for recessive inheritance [13,16].

PMP22, as a transmembrane glycoprotein with a four transmembrane (TM) domain, two extracellular (EC) loops, and three intracellular (IC) sequences including N- and C terminals, is mainly expressed in Schwann cells. It consists of 2–5% of the total protein in peripheral compact myelin that provides the electrical insulation of axons and has an essential role in the formation and maintenance of compact myelin [17]. A rat model with *PMP22* duplication showed CMT1A-like phenotypes, such as gait abnormalities, Schwann cell hypertrophy, and muscle weakness [18]. Homozygous mouse models with *Pmp22* point mutations exhibited a more severe peripheral myelin deficiency compared to heterozygous *Pmp22* knockout mice, suggesting gain-of-function or dominant-negative mechanisms of the mutant alleles [19].

This study was performed to identify pathogenic mutations in the *PMP22* gene in Korean patients with CMT by WES or targeted gene panel sequencing. As a result, we identified 14 pathogenic or likely pathogenic variants in *PMP22* from 21 independent CMT patients and families. Of these, we have previously reported five *PMP22* mutations in six Korean IPN families [20,21,22,23,24]. This study also analyzed the clinical phenotypic features and genotype–phenotype correlation for the patients with *PMP22* mutations.

## 2. Materials and Methods

### 2.1. Patients and Ethics Statements

This study was conducted in a cohort of 1,243 Korean CMT families. By examination of the copy numbers in the 17p12 region, 393 families were determined to have *PMP22* duplication. The remaining 850 CMT families were further investigated to detect *PMP22* small indels and point mutations. This study was approved by the Institutional Review Boards of Sungkyunkwan University, Samsung Medical Center (2018-05-102-002), and Kongju National University (KNU_IRB_2018-62). All participants were recruited from the Samsung Medical Center (Seoul, Korea) and provided written informed consent according to the Declaration of Helsinki. For minors involved in this study, written consent was provided by their parents.

### 2.2. Clinical and Electrophysiological Examinations

General clinical and electrophysiological information for affected individuals was obtained using the most recent data by the standard methods described by Park et al. [25]. To measure nerve conductions, the motor and sensory nerve conduction velocities and action potentials of the median, ulnar, peroneal, tibial, and sural nerves were determined. The age at onset was determined as when an individual initially noticed or showed distal dominant sensory-motor impairment. Physical disabilities were measured using several methods: functional disability scale (FDS), the CMT neuropathy score version 2 (CMTNS), Rasch-modified CMTNS (CMTNS-R), CMT examination score (CMTES), and Rasch-modified CMTES (CMTES-R) [26,27,28].

### 2.3. DNA Purification and Detection of Mutations

Genomic DNA was purified from peripheral blood using the HiGene Genomic DNA Prep Kit (Biofact, Daejeon, Korea). Proband samples with no 17p12 (*PMP22*) duplication or deletion were examined to find genetic causes by WES or targeted gene panel sequencing. Exome capture was performed using the SureSelect Human All Exon 50M Kit (Agilent Technologies, Santa Clara, CA, USA), and NGS was performed using the HiSeq 2000 or 2500 Genome Analyzer (Illumina, San Diego, CA, USA). Annotation and filtering of single nucleotide variants (SNVs) were performed by the methods of Park et al. with some alteration [25]. As the reference sequence, the UCSC assembly hg19 (GRCh37) was used (http://genome.ucsc.edu/ accessed on 1 May 2022). Rare variant frequencies were obtained from the 1000 Genomes Project (1000G; http://www.1000genomes.org/ accessed on 1 May 2022), the Genome Aggregation Database (gnomAD; https://gnomad.broadinstitute.org/ accessed on 1 May 2022), and the Korean Reference Genome Database (KRGDB; http://coda.nih.go.kr/coda/KRGDB/index.jsp/ accessed on 1 May 2022). Variant nomenclature followed the Human Genome Variation Society (HGVS) recommendations (https://www.hgvs.org/ accessed on 1 May 2022). Pathogenicity of the SNVs was evaluated and categorized into five grades following the American College of Medical Genetics and Genomics (ACMG) and American College of Pathology (AMP) guidelines (https://wintervar.wglab.org/ accessed on 1 May 2022). Pathogenic candidate variants were examined for all available family members by the Sanger sequencing method using the SeqStudio genetic analyzer (Life Technologies-Thermo Fisher Scientific, Foster City, CA, USA).

### 2.4. In Silico Prediction and Conservation Analysis

In silico analyses were performed to predict the mutation effect using the PROVEAN (http://provean.jcvi.org/ accessed on 15 April 2022), PolyPhen-2 (http://genetics.bwh.harvard.edu/pph2/ accessed on 15 April 2022), and MUpro (http://mupro.proteomics.ics.uci.edu/ accessed on 15 April 2022) programs. Conservation of the mutation sites and their flanking regions was analyzed using MEGA-X ver. 5.05 (http://www.megasoftware.net/ accessed on 15 April 2022). The genomic evolutionary rate profiling (GERP) score was determined by the GERP++ program (http://mendel.stanford.edu/SidowLab/downloads/gerp/ accessed on 15 April 2022). The secondary structure of single-strand DNA was predicted using the mFold algorithm (http://www.unafold.org accessed on 15 April 2022). Three-dimensional (3D) structures of wild-type and mutant PMP22 proteins were simulated by the I-TASSER program (https://zhanglab.ccmb.med.umich.edu/I-TASSER). The 3D structures were visualized using the Mol* feature of the Protein Data Bank (http://www.rcsb.org/ accessed on 15 April 2022).

### 2.5. Statistical Analysis

A non-parametric Kruskal–Wallis test for one-way ANOVA was used to compare the clinical characteristics of patients grouped according to the three types of CMT1 with different genetic causes. Missing clinical values were excluded from the analysis. Non-parametric Spearman r values were used to analyze the correlation between severity and age of onset. Differences were considered significant at *p* < 0.05. All statistical analyses and calculations were performed using GraphPad Prism ver. 8.00 (GraphPad Software, San Diego, CA, USA).

## 3. Results

### 3.1. Identification of Pathogenic Mutations in PMP22

We identified 14 dominant pathogenic or likely pathogenic *PMP22* mutations in 21 CMT families which included 72 participants (32 affected and 40 unaffected members) (Figure 1 and Table 1). All these mutations were unreported in the KRGDB and global databases of the 1000G, genomAD, and ESP. The identified mutations were confirmed for all involved family members by Sanger sequencing (Figure 2), and the genotypes for each mutation are provided below the examined individuals in Figure 1. Of these, four mutations (c.35A>G, c.179-1G>A, c.280_281delinsT, and c.319+1G>T) were novel, and another four mutations (c.47T>G, c.245T>C, c.318delT, and c.325T>C) were previously reported by our research group [20,21,22,23,24] but not reported in other population studies. De novo events were presumed in six mutations (c.35A>G, c.215C>T, c.245T>C, c.281delG, c.298G>A, and c.325T>C) involving nine families. For the families with de novo mutations, paternity was confirmed for the corresponding families with the PowerPlex Fusion System (Promega, Wisconsin-Madison, CA, USA).

The c.35A>G (p.H12R), as a novel mutation, was identified in a CMT1 male patient (FC618). His unaffected parents and younger sister did not have the same mutation, presumptively indicating a de novo event. The c.47T>G (p.L16R), which was once reported by Nam et al. [22], was found in a sporadic CMT1 male (FC541). His parents were considered to be unaffected by history taking, but genetic testing was not done. As a mutation of the same amino acid site, a p.L16P was reported in CMT1E with proptosis [29] and Dejerine-Sottas neuropathy patients [12]. The c.68C>G (p.T23R) mutation was found in three CMT1 families (FC50, FC303, and FC680). The FC50 and FC680 families showed an autosomal dominant inheritance, but the patient in the FC303 family was a sporadic case with both parents putatively unaffected (no genetic testing). This mutation has been reported twice including a Korean CMT1 case with deafness [7,30]. A novel c.179-1G>A splicing site mutation was found in a CMT1E family (FC970). In the same splicing acceptor site mutation, c.179-1G>C was reported in an HNPP family [31]. The c.215C>T (p.S72L) mutation was interestingly identified in five CMT1 families (FC285, FC376, FC732, FC829, and FC895). Moreover, four families (FC285, FC376, FC732, and FC829) showed a de novo event which was confirmed by the genetic testing of the father-mother-patient trios. This mutation has been reported many times as the underlying cause of CMT1 and DSS [10,11,21,23,24,32]. Two families (FC285 and FC829) were previously reported by our study group [21,23,24]. The c.245T>C (p.L82P), which was once reported by our study group [23], was identified in a CMT1 female patient (FC416). Her unaffected parents and elder brother did not have the same mutation, suggesting a de novo mutation. The c.256C>T (p.Q86X) stopgain mutation was identified in two CMT1 families (FC1102 and FC1325). In FC1102, the proband’s mother with the same mutation was also affected, while the proband in FC1325 was a sporadic case with unaffected parents (history taking). This mutation was reported twice by Numakura et al. and Liao et al. [33,34]. The c.281delG (p.G94Afs*16) frameshift mutation was identified in a CMT1 family (FC1061). It has been reported twice as an underlying cause of CMT1E [12,35]. The c.280_281delinsT (p.G94Sfs*16) mutation was identified in a sporadic female case with CMT1 (FC1088). It was an unreported novel mutation. The c.298G>A (p.G100R) was identified in a CMT1 female patient (FC1140). Her unaffected parents and elder brother did not have the same mutation, suggesting a de novo mutation. It was reported once as a pathogenic mutation of CMT1 [36]. The c.318delT (p.G107Vfs*3), which was reported once by our study group [20], was identified in a CMT1 family (FC35). His elder brother with the same mutation was also affected. Two affected brothers presumably inherited the mutation from their deceased affected father. Two mutations of c.319+1G>T and c.323T>C (p.L108P) were found in a sporadic CMT1 case, respectively. The c.319+1G>T splicing site mutation was unreported while a c.323T>C mutation was reported once as an underlying cause of Roussy–Levy syndrome (MIM 180800) which is considered to be a type of CMT1 [37]. The c.325T>C (p.C109R) mutation, which was reported by our study group [21,24], was identified in a CMT1 female patient (FC284). Her unaffected parents and younger brother did not have the same mutation, suggesting a de novo mutation.

### 3.2. In Silico Prediction, Conservation, and Conformational Changes

In silico analyses using the PolyPhen-2, PROVEAN, and MUpro programs predicted the pathogenicity for most of the missense mutations (Appendix A). The GERP scores were also calculated and were high values larger than 3.3 at all the missense mutation sites. Amino acids in most of the mutation sites and their flanking regions were highly conserved among vertebrate species from fish to mammals (Figure 3A). Except for two p.G94 frameshift mutations (p.G94Afs*16 and p.G94Sfs*16) positioning in the IC domain, the other mutations were located in the TM domains that are very important for the configuration of the protein interacting with the plasma membrane (Figure 3B).

The TM domains consisting of four α-helices are spatially associated with each other within the plasma membrane (Figure 4A). With the exception of p.T23, all missense mutations are positioned in the α-helices located in the TM domains. For the eight PMP22 mutant proteins with missense mutations, 3D conformational changes were predicted by the I-TASSER program (Figure 4B–I). It was suggested that the substitution of amino acids causes changes in the hydrogen bonds, affinity, or distance between the neighboring helixes. The conformational changes in the transmembrane regions could affect the functions and topology within the membrane.

### 3.3. Mutational Hotspots and an Atypical Case with Somatic Mutation

The c.68C>G and c.215C>T mutations that were located at the CpG sequences were particularly found in multiple families, suggesting mutational hotspots. Moreover, four families (FC285, FC376, FC732, and FC829) with the c.215C>T showed a de novo event. These mutations have been also reported several times in other studies [7,10,11,30,32]. These frequent mutations may be partly due to the putative palindromic sequences of the corresponding genomic DNA region forming a stem-loop hairpin (Figure 3C) and methylation in the CpG sites.

As an atypical case, the c.281delG (p.G94Afs*16) mutation identified in the FC1061 female patient (III-1) was also found in her father (II-1) but with low mutational loads: rates of 0.19, 0.23, and 0.32 in blood, oral swap, and hair root, respectively. Thus, the results suggest a somatic mutation in an early embryonic stage implicating genetic mosaicism. The proband showed a very severe clinical symptom with a very early infant onset (<1 year) and wheelchair bounding, while her father was clinically and electrophysiologically normal virtually in a recent checkup.

### 3.4. Prevalence of CMT1E in CMT Patients

The frequencies of CMT1E with *PMP22* mutations was calculated as 1.69% in the total examined CMT probands (*n* = 1243) and 2.47% in the CMT probands (*n* = 850) excluding CMT1A (Table 2). When the prevalence of total CMT patients was compared with other countries, it was roughly similar with China at 0.94–2.38% [38,39], Italy at 1.36% [40], and Brazil at 2.10% [41] while it was higher than other examined populations, such as in Hungary at 0.37% [42] and Spain at 0.46% [43]. The prevalence of CMT1E in Japanese CMT patients excluding CMT1A ranged from 0.81–1.29%, which is a considerably low frequency compared with that of Korean CMT patients [44,45].

### 3.5. Characterization of Clinical and Electrophysiological Phenotypes

We analyzed the clinical and electrophysiological phenotypes of the CMT1E patients with *PMP22* mutations. Major clinical features are provided in Table 3. Detailed clinical and electrophysiological data are provided in Appendix A, respectively. Mean onset ages for all the examined CMT1E patients were measured to be 12.8 ± 11.1 years. As the physical disability values, mean CMTNS, CMT-E, CMTES, CMTES-R, and FDS scores were measured as 21.1 ± 8.1, 25.8 ± 9.0, 16.2 ± 5.6, 19.7 ± 6.7, and 3.8 ± 2.4, respectively. Mean median motor nerve conduction velocity (MNCV) was measured to be 11.5 ± 13.0 m/s in CMT1E patients. Most patients showed foot deformity (84.4%), and hearing loss was diagnosed in 40.6% of the patients (13 of 32).

The mean onset age of the CMT1E patients showed no significant difference with those of CMT1A with *PMP22* duplication (10.9 ± 5.8) [46] and CMT1B with *MPZ* mutations (9.3 ± 10.7) (Figure 5A) [46,47]. However, the distribution of onset ages showed a noticeable difference between the CMT1 groups. Infancy onset (<1 year) was only observed in CMT1E with a high frequency of 19.4% (6/32) while the adult-onset (≥19 years) frequency of 37.5% (12/32) in CMT1E was higher than those of CMT1A (5.6%) and CMT1B (14.6%). The CMT1E group roughly showed a U-shaped distribution compared to the bell-shaped distribution of CMT1A and the L-shaped distribution of CMT1B (Figure 5B).

The mean CMTNS score (21.1 ± 8.1) was significantly higher than that of CMT1A (10.2 ± 4.3, *p* < 0.0001) while it was not significantly different with that of CMT1B (16.9 ± 6.5, *p* = 0.408) (Figure 5C). For the FDS, the mean value of CMT1E (3.8 ± 2.4) was also significantly higher than that of CMT1A (1.6 ± 0.8, *p* < 0.0001), but it was similar with that of CMT1B (2.8 ± 1.3, *p* = 0.999) (Figure 5D). The mean median MNCV of CMT1E (11.5 ± 13.0 m/s) was significantly slower than that of CMT1A (18.2 ± 4.9 m/s, *p* = 0.0006) or similar with that of CMT1B (10.9 ± 9.7 m/s, *p* = 0.999) (Figure 5E) [46,47].

As an important feature of the clinical phenotypes in CMT1E patients, a clear correlation between early onset with severe symptoms and late onset with mild symptoms was observed (Figure 6). That is, the physical disability scores measured by CMTNS and FDS showed clearly negative correlations by increasing onset ages (CMTNS: *r* = −0.595, *p* = 0.0007; FDS: *r* = −0.644, *p* < 0.0001) (Figure 6A,B). The median MNCV, as a major electrophysiological value, showed a positive correlation by increasing onset ages *(r* = 0.546, *p* = 0.002) (Figure 6C).

## 4. Discussion

We identified 14 pathogenic or likely pathogenic point mutations in *PMP22* from 21 CMT families or sporadic cases in a Korean CMT cohort genetic study. Of these, four mutations (c.35A>G, c.179-1G>A, c.280_281delinsT, and c.319+1G>T) were unreported novel variants. Another four mutations (c.47T>G, c.245T>C, c.318delT, and c.325T>C) were reported by our research group [20,21,22,23,24] but not reported in other populations. Most mutations were located in the well-conserved TM domains and predicted to be pathogenic at least by one or more in silico analyses. All the mutations were evaluated as pathogenic or likely pathogenic by the ACMG/AMP guidelines.

The prevalence of CMT1E patients with *PMP22* mutations was determined to be 1.69% in the total independent patients and 2.47% in the CMT patients excluding CMT1A. This CMT1E prevalence corresponds to the third most common subtype of demyelinated CMT1 after CMT1A with *PMP22* duplication and CMT1B with *MPZ* mutations [47]. The prevalence of CMT1E showed a difference according to the countries studied, but there appears to be no difference by continent. The prevalence of CMT1E in Korean CMT patients was similar to that of Chinese patients [38,39], but it was significantly higher than that of Japanese CMT patients without CMT1A [44,45]. When the prevalence was compared with European populations, it was clearly higher than those of Hungarians [42] and Spaniards [43]. However, it was slightly higher than that of Italians. In Brazilians, the frequency in total patients was similar or slightly higher compared with East Asians, but 7.41% in patients excluding CMT1A is the highest value worldwide [41].

Of note, high frequencies of de novo mutations were observed at a rate of 42.9% in the examined families. The rate increases to over 60% when counting father-mother-child trio families only. Although *PMP22* de novo mutations have been frequently reported by other studies [9,15], the rate was higher than those shown in the other CMT gene mutations in Korea compared to the 45.0% and 18.7% of the trio families with the *MPZ* mutations and *PMP22* duplication causing CMT1A [47], respectively. Only two and one de novo cases were observed in 11 families with small heat shock protein (sHSP) gene mutations [48] and 13 families with aminoacyl tRNA-synthetase (ARS) gene mutations [49]. In particular, c.215C>T (p.S72L) de novo mutation, which was observed in 4 of 5 families with the corresponding mutation, is suggested to locate in a mutational hotspot. The c.215C nucleotide is positioned in the single-stranded DNA region forming a stem-loop hairpin. In addition, the c.215C was reported as a strong methylation site in the UCSC Genome Browser (http://genome.ucsc.edu/ accessed on 1 May 2022).

As an important feature of the clinical phenotypes, the CMT1E patients showed a clear correlation between early onset with severe symptoms and late onset with mild symptoms. This aspect is similar to the clinical features of CMT2A patients with *MFN2* mutations [50]. More than 40% of CMT1E patients showed hearing loss, which is consistent with previous studies [5,6,7,30]. Regarding onset, more than 1/3 of all CMT1E patients examined in this study showed early onset at the age of 5 years or younger, whereas the rate of patients with late onset after 19 years old was also high with more than 37% which roughly showed a dimorphic distribution. This distribution showed a clear difference from the CMT1A patients with 15% and 5% and CMT1B patients with 52% and 15% of the approximate ratio of early onset and late onset. However, the mean onset age of the CMT1E patients was similar to those of CMT1A and CMT1B [46,47]. When the physical and electrophysiological severities of CMT1E were compared with those of CMT1A and CMT1B in terms of CMTNS, FDS, and MNCV, CMT1E was significantly more severe than CMT1A while showing similar symptoms with CMT1B.

More than 100 genes are associated with CMT in variable inheriting modes. Among them, *PMP22* defects are found in more than one-third of CMT patients, and the genomic dosage of *PMP2*2 is often the subject of prenatal diagnosis including the preimplantation genetic diagnosis [51]. In particular, the inhibitory effects of *PMP22* overexpression by treatment of ascorbic acid, progesterone antagonist, and siRNA have suggested personalized therapeutic strategies by genetic causes [52,53,54]. Therefore, data for *PMP22* mutations could be applied to the early diagnosis and treatment of CMT patients.

## 5. Conclusions

This genetic cohort study identified 14 *PMP22* mutations in 21 Korean families as the underlying causes of CMT1E phenotypes. Of these, eight mutations were not reported in other countries. We carefully analyzed the genotype–phenotype correlations and compared the clinical phenotypes with other frequent CMT1 types of CMT1A and CMT1B. This study particularly provided detailed physical and electrophysiological data for all the examined patients. We believe that our results will be useful for the reference data of Koreans and the molecular diagnosis of CMT1 with or without deafness.

## Figures and Tables

**Figure 1 genes-13-01219-f001:**
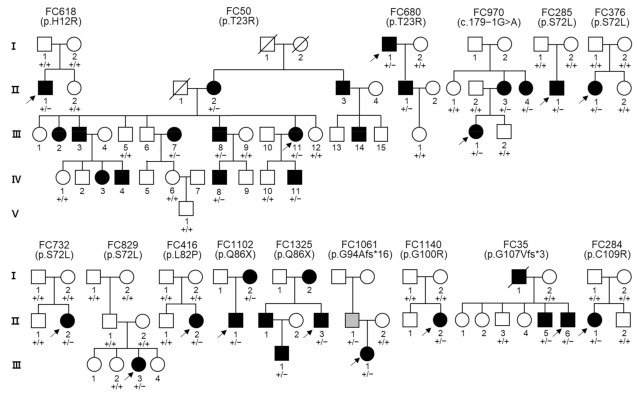
Charcot-Marie-Tooth disease type 1E families with *PMP22* mutations. Genotypes of the *PMP22* mutations are provided at the bottom of all examined individuals. Arrows indicate the proband and roman numerals at the left side indicate generation within each pedigree (unfilled symbols (□, ○): unaffected individuals; black-filled symbols (■, ●): affected individuals; gray-filled symbol: no symptomatic individual having mutation with mosaic condition; +: wild type allele; −: mutant allele).

**Figure 2 genes-13-01219-f002:**
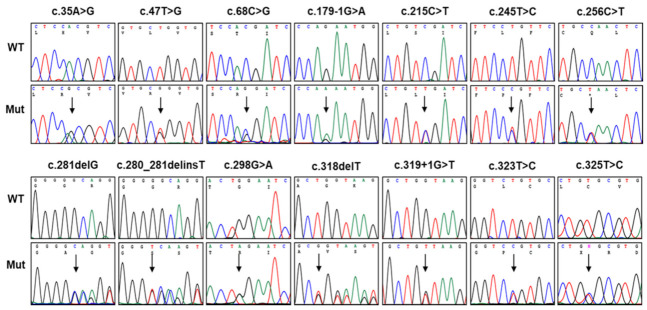
Chromatopherograms of the *PMP22* mutations. They were obtained by Sanger sequencing, and the mutation sites are indicated by vertical arrows (WT: wild-type allele, Mut: mutant allele).

**Figure 3 genes-13-01219-f003:**
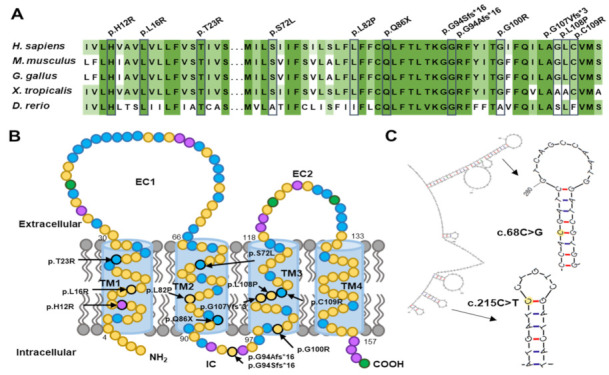
Conservation and location of mutation sites. (**A**) Conservation among species. Reference sequences are from NP_000295.1 (*Homo sapiens*), NP_001289184.1 (*Mus musculus*), NP_001264000.1 (*Gallus gallus*), NP_001025552.1 (*Xenopus tropicalis*), and NP_958468.1 (*Danio rerio*). (**B**) Location of mutations in a schematic PMP22 protein (EC: extracellular domain. IC: intracellular domain; TM1~4: transmembrane domain 1~4). Nonpolar, polar, basic, and acidic amino acids are indicated in yellow, blue, violet, and green, respectively. (**C**) Predicted secondary structures of single-strand DNA for flanking regions of c.68C>G and c.215C>T. Two nucleotides are indicated in yellow.

**Figure 4 genes-13-01219-f004:**
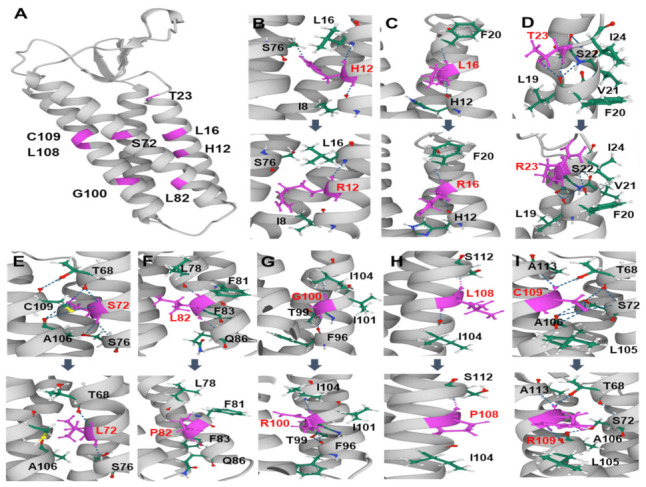
Predicted conformational changes of the PMP22 protein by mutations. The eight mutation sites are indicated in pink. Hydrogen bonds are indicated by blue dotted lines, and carbon, hydrogen, nitrogen, and oxygen are indicated in green, white, blue, and red, respectively. (**A**) Whole structure of the wild PMP22. (**B**–**I**) Predicted conformational changes in the surrounding of the mutated residues. In each panel, the wild structure and mutant are arrayed top and bottom, respectively (**B**: p.H12R, **C**: p.L16R, **D**: p.T23R, **E**: p.S72L, **F**: p.L82P, **G**: p.G100R, **H**: L108P, and **I**: p.C109R).

**Figure 5 genes-13-01219-f005:**
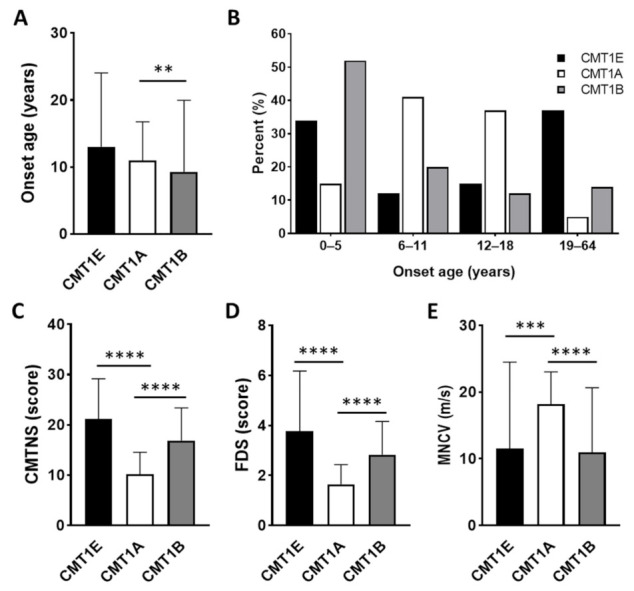
Comparison of clinical phenotypes among the Charcot-Marie-Tooth disease type 1 groups of CMT1E, CMT1A, and CMT1B (**: *p* ≤ 0.01, ***: *p* ≤ 0.001, and ****: *p* ≤ 0.0001). (**A**) Comparison of onset ages. (**B**) Distribution of patients by onset ages. (**C**) Comparison of CMTNS. (**D**) Comparison of FDS. (**E**) Comparison of median MNCV.

**Figure 6 genes-13-01219-f006:**
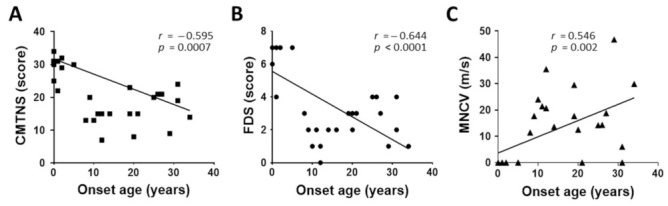
Correlations of clinical and electrophysiological phenotypes with onset ages. (**A**) CMTNS vs. onset age, (**B**) FDS vs. onset age, and (**C**) MNCV vs. onset age.

**Table 1 genes-13-01219-t001:** *PMP22* mutations in Charcot-Marie-Tooth disease type 1E patients.

Mutations ^1^		No. of Families	Family IDs	Mutant Allele Frequencies ^2^			ACMG-AMP	Notes and References
Nucleotide	Amino Acid			1000G	gnomAD	KRGDB		
c.35A>G	p.H12R	1	FC618	NR	NR	NR	P	Novel, de novo
c.47T>G	p.L16R	1	FC541	NR	NR	NR	LP	[22]
c.68C>G	p.T23R	3	FC50, FC303, FC680	NR	NR	NR	P	[7,30]
c.179-1G>A	Splicing site	2	FC970	NR	NR	NR	LP	Novel
c.215C>T	p.S72L	5	FC285, FC376, FC732, FC829, FC895	NR	NR	NR	P	[10,11,21,23,24,32], de novo: FC285, FC376, FC732, FC829
c.245T>C	p.L82P	1	FC416	NR	NR	NR	LP	[23], de novo
c.256C>T	p.Q86X	2	FC1102, FC1325	NR	NR	NR	P	[33,34]
c.281delG	p.G94Afs*16	1	FC1061	NR	NR	NR	P	[12,35], de novo
c.280_281delinsT	p.G94Sfs*16	1	FC1088	NR	NR	NR	LP	Novel
c.298G>A	p.G100R	1	FC1140	NR	NR	NR	P	[36], de novo
c.318delT	p.G107Vfs*3	1	FC35	NR	NR	NR	P	[20]
c.319+1G>T	Splicing site	1	FC608	NR	NR	NR	P	Novel
c.323T>C	p.L108P	1	FC1060	NR	NR	NR	P	[37]
c.325T>C	p.C109R	1	FC284	NR	NR	NR	P	[21,24], de novo

^1^ Reference nucleotide and amino acid sequences: NM_000304.4 and NP_000295.1. ^2^ Minor allele frequencies were obtained from the 1000 Genomes Project (1000G), the Genome Aggregation Database (gnomAD), the Exome Sequencing Project (ESP), and Korean Reference Genome Database (KRGDB). Abbreviations: ACMG-AMP: the American College of Medical Genetics and Genomics-American College of Pathology, CMT1E: Charcot-Marie-Tooth disease type 1E, LP: likely pathogenic, NR: nonreported, P, pathogenic.

**Table 2 genes-13-01219-t002:** Prevalence of Charcot-Marie-Tooth disease type 1E patients with *PMP22* mutations.

Populations	Examined Numbers		Number of CMT1E Patients	Prevalence (%)		References
	Total	CMT1A Exclusion		Total	CMT1A Exclusion	
Korean	1243	850	21	1.69	2.47	This study
South Chinese	421	336	10	2.38	2.98	[39]
Chinese (Taiwan)	427	219	4	0.94	1.83	[38]
Japanese	-	2598	21	-	0.81	[45]
Japanese	-	1005	13	-	1.29	[44]
Italian	295	238	4	1.36	1.68	[40]
Brazilian	286	81	6	2.10	7.41	[41]
Hungarian	531	320	2	0.37	0.63	[42]
Spanish	438	254	2	0.46	0.79	[43]

**Table 3 genes-13-01219-t003:** Clinical and electrophysiological features of Charcot-Marie-Tooth disease type 1E patient.

Patient ID	Sex	Age (yrs)		CMTNS	CMTES	FDS	MRC ^1^		DTR ^2^ Knee/Ankle	HL	Median Nerve Conduction Study			
		Exam.	Onset				UEx	LEx			CMAP (mV)	MNCV (m/s)	SNAP (μV)	SNCV (m/s)
FC618 (II-1)	M	1	1	31	23	7	+++	+++	A/A	No	A	A	A	A
FC541	M	13	12	ND	ND	1	+	++	A/A	Yes	ND	ND	ND	ND
FC50 (II-2)	F	84	31	24	20	4	++	++	A/A	Yes	A	A	A	A
FC50 (III-7)	F	64	25	20	16	4	++	+++	A/A	Yes	0.7	14.2	A	A
FC50 (III-8)	M	49	19	23	18	3	++	++	A/A	Yes	1.2	17.7	A	A
FC50 (III-11)	F	45	26	21	17	4	++	+++	A/A	Yes	4.1	14.3	A	A
FC50 (IV-8)	M	20	12	15	11	1	+	++	A/A	Yes	6.1	20.7	A	A
FC50 (IV-11)	M	15	11	15	11	2	+	++	A/A	Yes	8.9	21.4	A	A
FC303	M	20	10	13	9	1	+	+	D/A	No	11.6	24	A	A
FC680 (I-1)	M	70	27	21	17	3	++	++	A/A	Yes	8.1	18.8	2.4	18.2
FC680 (II-1)	M	34	9	20	16	2	+	+	A/A	Yes	3.3	17.7	A	A
FC970 (II-3)	F	40	29	9	9	1	+	+	D/D	No	19.0	46.8	28.6	35.3
FC970 (II-4)	F	39	34	14	11	1	+	+	D/D	Yes	12.3	29.9	9.4	26.2
FC970 (III-1)	F	13	12	7	6	0	-	+	N/D	No	17.4	35.5	20.9	29.2
FC285 (II-1)	M	16	<1	31	23	7	+++	+++	A/A	No	A	A	A	A
FC376 (II-1)	F	11	2	32	24	7	+++	+++	A/A	No	A	A	A	A
FC732 (II-2)	F	7	2	29	21	7	++	+++	A/A	No	A	A	A	A
FC829 (III-3)	F	12	<1	25	17	7	++	+++	A/A	No	A	A	1.8	20.8
FC895	M	7	<1	31	23	6	+++	+++	A/A	No	A	A	A	A
FC416 (II-2)	F	28	<1	30	22	7	++	+++	A/A	No	A	A	A	A
FC1102 (I-2)	F	47	16	ND	ND	2	+	++	D/A	No	ND	ND	ND	ND
FC1102 (II-1)	M	21	14	15	11	2	+	++	D/A	No	6.3	13.6	A	A
FC1325 (II-3)	M	54	8	13	15	3	+	++	A/A	Yes	1.8	11.4	A	A
FC1325 (III-1)	M	31	20	8	11	2	+	+	D/A	No	6.8	12.5	A	A
FC1061 (III-1)	F	26	<1	31	23	7	+++	+++	A/A	No	A	A	A	A
FC1088	F	48	5	30	22	7	+++	+++	A/A	No	A	A	A	A
FC1140 (II-2)	F	6	1	22	16	4	++	+++	A/A	No	A	A	A	A
FC35 (II-5)	M	34	20	ND	ND	3	+	++	A/A	Yes	ND	ND	ND	ND
FC35 (II-6)	M	32	19	15	12	3	+	++	A/A	Yes	10.0	29.5	A	A
FC608	M	39	31	19	12	2	+	++	A/A	No	0.5	6.0	A	A
FC1060	M	33	21	15	9	3	+	++	D/A	No	A	A	A	A
FC284 (II-1)	F	15	<1	34	26	7	+++	+++	A/A	No	A	A	A	A

^1^ Muscle weakness in upper limbs (UEx): -: normal, +: intrinsic hand weakness of 4/5 on the Medical Research Council (MRC) scale, ++: intrinsic hand weakness < 4/5 on the MRC scale, +++: proximal weakness. Muscle weakness in lower limbs (LEx): +: ankle dorsiflexion of 4/5 on the MRC scale, ++: ankle dorsiflexion < 4/5, +++: proximal weakness. ^2^ Deep tendon reflex (DTR): A: absent, D: diminished, N: normal reflex. Abbreviations: A: absent action potential, CMAP: compound muscle action potential, CMTNS: Charcot-Marie-Tooth neuropathy score, CMTES: CMT examination score, HL: hearing loss, MNCV: motor nerve conduction velocity, ND: not determined, SNAP: sensory nerve action potential, SNCV: sensory nerve conduction velocity.

## Data Availability

The data presented in this study are available in the article. Additional WES and TS data are available upon request to the corresponding authors.

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
