# Peer review of "Peripheral Myelin Protein 22 Gene Mutations in Charcot-Marie-Tooth Disease Type 1E Patients"

_genes, 2022, doi:10.3390/genes13071219_

Round 1

Reviewer 1 Report

[Genes] Manuscript ID: genes-1778264

Peripheral Myelin Protein 22 Gene Mutations in Charcot-Marie-Tooth

Disease Type 1E Patients

In the methods the authors state that NCSs of the median, ulnar, peroneal, tibial, and sural nerves were performed.

However, only results of the median nereve were presented

Results of the other nerves should also be shown particularlaly those of the lower limb nerves

6/22

Author Response

Thank you very much for your comments. As stated in the Materials and Methods, we measured median, ulnar, peroneal, and tibial nerves for the motor nerve conduction and median, ulnar, and sural nerves for the sensory nerve conduction. Motor compound muscle action potential (CMAP), motor nerve conduction velocity (MNCV), sensory nerve action potential (SNAP), and sensory nerve conduction velocity (SNCV) were presented in the main text (Table 3), while other nerve conduction data are provided in supplementary Table S3 due to large dataset.

Reviewer 2 Report

This manuscript is a very interesting paper in which the authors identified several peripheral myelin protein 22 (PMP22) mutations in Korean families as the underlying causes of Charcot-Marie-Tooth disease type 1E (CMT1E). Given that Charcot-Marie-Tooth disease is the most commonly inherited peripheral neuropathy affecting more than 2.8 million people worldwide, this study could be very useful to establish the early diagnosis of this disorder.

The manuscript is good and well organized. However, the introduction Section could be improved. For example, the authors could briefly discuss about the prenatal diagnosis of this disorder, and if is there a treatment for CMT or if there are currently no treatments for this disorder?

Author Response

Thank you very much for your comments. As the comments, following sentences were added in the Discussion section instead of Introduction: More than 100 genes are associated with CMT in variable inheriting modes. Among them, PMP22 defects are found in more than one third of CMT patients, and the genomic dosage of PMP22 is often the subject of the prenatal diagnosis including the preimplantation genetic diagnosis [51]. In particular, the inhibitory effects of PMP22 overexpression by treatment of ascorbic acid, progesterone antagonist, and siRNA have suggested personalized therapeutic strategies by genetic causes [52-54]. Therefore, data for PMP22 mutations could be applied to the early diagnosis and treatment of CMT patients.